# Butyrate Conversion by Sulfate-Reducing and Methanogenic Communities from Anoxic Sediments of Aarhus Bay, Denmark

**DOI:** 10.3390/microorganisms8040606

**Published:** 2020-04-22

**Authors:** Derya Ozuolmez, Elisha K. Moore, Ellen C. Hopmans, Jaap S. Sinninghe Damsté, Alfons J. M. Stams, Caroline M. Plugge

**Affiliations:** 1Laboratory of Microbiology, Wageningen University & Research, 6708 WE Wageningen, The Netherlands; adaderya@gmail.com (D.O.); fons.stams@wur.nl (A.J.M.S.); 2Department of Marine Microbiology and Biogeochemistry, NIOZ Royal Netherlands Institute for Sea Research, and Utrecht University, P.O. Box 59, 1790 AB Den Burg, The Netherlands; mooreek@rowan.edu (E.K.M.); Ellen.Hopmans@nioz.nl (E.C.H.); jaap.damste@nioz.nl (J.S.S.D.); 3Department of Environmental Science, Rowan University, 201 Mullica Hill Rd, Glassboro, NJ 08028, USA; 4Faculty of Geosciences, Utrecht University, P.O. Box 80.021, 3508 TA Utrecht, The Netherlands

**Keywords:** marine sediment, sulfate-reducing bacteria, methanogenic archaea, syntrophy, intact polar lipids, Aarhus Bay

## Abstract

The conventional perception that the zone of sulfate reduction and methanogenesis are separated in high- and low-sulfate-containing marine sediments has recently been changed by studies demonstrating their co-occurrence in sediments. The presence of methanogens was linked to the presence of substrates that are not used by sulfate reducers. In the current study, we hypothesized that both groups can co-exist, consuming common substrates (H_2_ and/or acetate) in sediments. We enriched butyrate-degrading communities in sediment slurries originating from the sulfate, sulfate–methane transition, and methane zone of Aarhus Bay, Denmark. Sulfate was added at different concentrations (0, 3, 20 mM), and the slurries were incubated at 10 °C and 25 °C. During butyrate conversion, sulfate reduction and methanogenesis occurred simultaneously. The syntrophic butyrate degrader *Syntrophomonas* was enriched both in sulfate-amended and in sulfate-free slurries, indicating the occurrence of syntrophic conversions at both conditions. Archaeal community analysis revealed a dominance of *Methanomicrobiaceae*. The acetoclastic *Methanosaetaceae* reached high relative abundance in the absence of sulfate, while presence of acetoclastic *Methanosarcinaceae* was independent of the sulfate concentration, temperature, and the initial zone of the sediment. This study shows that there is no vertical separation of sulfate reducers, syntrophs, and methanogens in the sediment and that they all participate in the conversion of butyrate.

## 1. Introduction

Coastal marine ecosystems receive a regular input of organic matter from primary production of plankton, macroalgae, and vascular plants and the influx of rivers, and they are important in the remineralization of organic matter [1,2]. Most particulate organic matter is rapidly deposited on the coastal shelf. High microbial activity in the sediment layers leads to the formation of distinct biogeochemical zones. The depth range of each zone varies strongly depending on chemical changes in the sediment pore water, the rates of sediment accumulation, and replenishment of electron acceptors from overlying seawater [3]. In coastal marine sediments, the thickness of the oxic surface layer can be just a few millimeters [2]. Where oxygen is depleted, the sediment becomes anoxic. In the anoxic part of the sediment, nitrate, manganese, iron, sulfate, and carbon dioxide, in an order of decreasing energy gain, serve as terminal electron acceptors for the mineralization processes [4,5].

Anaerobic degradation of organic matter in marine sediments is a complex sequential process in which a variety of physiologically different microorganisms take part [1]. The first step is an extracellular hydrolysis of polymers (polysaccharides, proteins, nucleic acids, and lipids). Primary fermentative bacteria ferment the monomers and oligomers to fatty acids, alcohols, aromatic acids, H_2,_ and CO_2_. Some of these fermentation products, such as acetate, H_2_, CO_2_, and other one-carbon compounds, can be converted directly to methane and carbon dioxide by methanogens. In methanogenic environments, secondary fermenters or proton reducers convert alcohols, aliphatic and aromatic fatty acids to acetate, formate, H_2,_ and CO_2_, which are subsequently used by the methanogens [6]. In this way, a syntrophic relationship is established between microorganisms that convert organic compounds and methanogens (Appendix A) [6]. The conversion of polymers in sulfate-rich anoxic habitats such as marine sediments is different. In contrast to methanogens, sulfate-reducing bacteria can metabolize all products of primary fermentation and oxidize them to carbon dioxide, while reducing sulfate to sulfide [7].

Sulfate reduction and methanogenesis have been reported to occur simultaneously in anoxic marine environments where input of organic carbon is high [3,8,9]. In such environments, the use of noncompetitive methylated substrates by [10] methanogens was first suggested to enable cohabitation of both functional microbial groups. However, several studies demonstrated the consumption of common substrates, H_2_ and acetate, by both microbial groups in sulfate-rich sediments [11]. Sulfate-reducing bacteria (SRB) were detected in the methane zone in comparable numbers as in the sulfate zone of sediments of the Black Sea and Aarhus Bay [12,13]. The niche differentiation of the two groups of microbes is not fully understood. Thus far, syntrophic conversion of fatty acids in marine environments has received little attention. In [14], marine butyrate-degrading syntrophs in the sulfate zone were detected and it was suggested that syntrophic interactions constitute a significant methane source in marine sediments. As high methane concentrations were observed under conditions with high sulfate concentrations in a hydrocarbon-contaminated aquifer, it was concluded that butyrate was metabolized mainly syntrophically [15]. Other studies reported the existence of butyrate-degrading syntrophic genera *Syntrophus* and *Syntrophomonas* in different sulfate-containing environments [16,17,18].

As both *Syntrophus* and *Syntrophomonas* cannot reduce sulfate, and known butyrate-degrading SRB are not able to grow syntrophically [19], it is not clear if butyrate-degrading SRB can perform a syntrophic lifestyle to enable them to thrive in high- and low-sulfate environments. Thus, it is important to understand how sulfate reducers and syntrophs interact with methanogens in the presence and absence of sulfate.

In this study, we investigated butyrate-degrading communities from the sulfate, sulfate–methane transition, and methane zone in anoxic sediments of Aarhus Bay, Denmark. We set up batch slurry incubations and applied different sulfate concentrations (0, 3, 20 mM) to see which butyrate-utilizing sulfate reducers and syntrophs become dominant in response to sulfate availability and which methanogens contribute to butyrate conversion in the presence or absence of sulfate.

## 2. Materials and Methods

### 2.1. Sediment Sampling

Sediment was collected at station M1 located in the central part of the Aarhus Bay, Denmark (56°07′066″ N, 10°20′793″ E) at a water depth of 15 m during a research cruise with RV Tyra in May 2011. The in situ temperature of the sediment was ~9 °C. Two 3 m-long gravity cores were retrieved; one of them was sectioned in 10 cm depth intervals for physical, chemical, and molecular analyses and the other one was kept intact in the core liners, in sealed gas-tight plastic bags containing AnaeroGen sachets (Oxoid, Landsmeer, Netherlands) at 4 °C until further processing.

### 2.2. Sediment Pore Water Analysis

Methane, sulfate, and sulfide analysis from sediment pore water was performed on the sampling day at the laboratories of Center for Geomicrobiology, Aarhus University, Denmark. The pore water concentrations of sulfate and methane were determined as described by [20]. Hydrogen sulfide was quantified in zinc-preserved pore water samples by the methylene blue method [21].

### 2.3. Sediment Slurry Incubations

Sediments from three different biogeochemical zones were used to establish replicate sediment slurries. Stored sediment cores were processed under aseptic and anaerobic conditions in the laboratory. Subsamples were mixed in an anaerobic chamber and used as inoculum for sediment slurry enrichments. An amount of 100 mL of the homogenized sediment from each zone was mixed with 300 mL of anaerobic mineral salts medium in 1 L serum bottles. The medium composition was as follows (g/L): KH_2_PO_4_ (0.41), Na_2_HPO_4_·2H_2_O (0.53), NH_4_Cl (0.3), CaCl_2_·2H_2_O (0.11), MgCl_2_·6H_2_O (3), NaHCO_3_ (4), Na_2_S·9H_2_O (0.024), KCL (0.5), NaCl (25). The medium was supplemented with 1 mL/L of acid trace element solution (50 mM HCl, 1 mM H_3_BO_3_, 0.5 mM MnCl_2_, 7.5 mM FeCl_2_, 0.5 mM CoCl_2_, 0.1 mM NiCl_2_, 0.5 mM ZnCl_2_), 1 mL/L of alkaline trace element solution (10 mM NaOH, 0.1 mM Na_2_SeO_3_, 0.1 mM Na_2_WO_4_, 0.1 mM Na_2_MoO_4_), and 10 mL/L vitamin solution (Biotin 20 mg/L, Nicotinamid 200 mg/L, p-Aminobenzoic acid 100 mg/L, Thiamin 200 mg/L, Pantothenic acid 100 mg/L, Pyridoxamine 500 mg/L, Cyanocobalamine 100 mg/L, Riboflavin 100 mg/L). Bottles were closed with butyl rubber stoppers, and the headspace was exchanged with N_2_/CO_2_ (80:20%, *v*/*v*), and the final pressure was adjusted to 1.5 kPa. An amount of 10 mM Na-butyrate was used as carbon source with and without 20 mM sulfate in sulfate zone and methane zone slurries, and with 3 mM and 20 mM sulfate for sulfate–methane transition zone slurries. Control bottles were prepared in the same manner, without addition of butyrate. One set of the bottles representing each condition in duplicate was incubated at 10 °C to mimic in situ sediment temperature [22], and the other set was kept at 25 °C statically throughout the experiment. Regular liquid and gas sampling was performed to monitor substrate consumption, product formation, and to carry out molecular analysis. Regular additions of butyrate and/or sulfate were done as soon as they were depleted to maintain the slurry conditions.

### 2.4. Analytical Methods

CH_4_ in the headspace of slurries was analyzed by gas chromatography as described previously [23]. Volatile fatty acids from centrifuged (10,000× *g*, 10 min) samples of the sediment slurries were analyzed by HPLC as described previously [23]. Data analyses were performed using ChromQuest (Thermo Scientific, Waltham, MA, USA) and Chromeleon software (Thermo Scientific, Waltham, MA). Sulfate and sulfide were quantified as described in [24]. 

### 2.5. DNA Extraction

Genomic DNA was extracted from the sediment and enrichment slurry samples using the FastDNA SPIN Kit for Soil (MP Biomedicals, Solon, OH, USA) according to manufacturer’s protocol, with the following adaptations to increase the DNA yield. Sediment or slurry sample (5 mL) was suspended in 10 mL of phosphate-buffered saline (PBS), sonicated at low power to detach cells from the solid phase, and was centrifuged at 4700× *g* for 20 min. The supernatant was discarded and remaining pellet was re-suspended in 10 mL 0.5 M EDTA, pH 8, and incubated overnight at 4 °C to dissolve humic substances. After incubation, the suspension was centrifuged at 4700× *g* for 10 min, washed with PBS, and DNA was extracted from the pellet. The DNA was quantified with a Nanodrop ND-1000 spectrophotometer (Nanodrop Technologies, Wilmington, DE, USA).

### 2.6. 16S rRNA Gene Amplicon Pyrosequencing

Bacterial 16S rRNA gene fragments were amplified using barcoded primers covering the V1–V2 region of the bacterial 16S rRNA gene. The forward primer consisted of the 27F-DegS primer (5′-GTTYGATYMTGGCTCAG-3′) appended with the titanium sequencing adaptor A (5′-CCATCTCATCCCTGCGTGTCTCCGACTCAG-3′) and an eight nucleotide sample specific barcode at the 5′ end. An equimolar mix of two reverse primers was used, i.e., 338RI (5′-GCWGCCTCCCGTAGGAGT-3′) and 338RII (5′-GCWGCCACCCGTAGG TGT-3′) that carried the titanium adaptor B (5′-CCTATCCCCTGTGTGCCTTGGCAG TCTCAG-3′) at the 5′ end. Sequences of both titanium adaptors were purchased from GATC Biotech (Konstanz, Germany). Genomic DNA was diluted to a concentration of 20 ng/µL based on Qubit^®^ 2.0 fluorometer readings, and amplicons were generated as described in [25]. Amplicons were sequenced using a 454 FLX genome sequencer in combination with titanium chemistry (GATC Biotech AG, Konstanz, Germany. The demultiplexed reads of the 16S rRNA gene amplicon sequences of *Bacteria* were deposited at the European Nucleotide Archive (ENA) under study PRJEB36640.

### 2.7. Analysis and Interpretation of the Pyrosequencing Data

Pyrosequencing data was analyzed as described in [24]. The relative amount of reads of every operational taxonomic unit (OTU) to the total amount of reads per sample was quantified, and the average relative amount of reads per representative OTU of each slurry sample was calculated.

### 2.8. Illumina HiSeq Analysis of Archaeal Community

Extracted DNA from the samples taken on the last incubation day from all slurries was used for archaeal community analysis. Barcoded amplicons were generated using a two-step PCR method that was shown to reduce the impact of barcoded primers on the outcome of microbial profiling [26]. Amplification was done as described by [27].

16S rRNA gene sequencing data was analyzed using NG-Tax, an in-house pipeline [28]. Paired-end libraries were filtered to contain only read pairs with perfectly matching barcodes, and those barcodes were used to demultiplex reads by sample. Operational taxonomic units (OTUs) were defined using an open reference approach, and taxonomy was assigned to those OTUs using the SILVA 16S rRNA gene reference database [29]. Microbial composition plots were generated using a workflow based on Quantitative Insights Into Microbial Ecology (QIIME) v1.2 [30]. The demultiplexed reads of the 16S rRNA gene amplicon sequences of the *Archaea* were deposited at the European Nucleotide Archive (ENA) under study PRJEB36882.

### 2.9. Intact Polar Lipid (IPL) Extraction and HPLC-ESI/IT/MS Analysis

The slurry samples which were used for molecular analysis were also used for lipid analysis. Additionally, methane zone slurry samples which were not analyzed for molecular analysis were extracted and analyzed in order to confirm the presence or absence of trimethylornithine lipids (TMOs). IPLs were extracted from ~0.1 g freeze-dried powdered samples using a modified Bligh and Dyer method [31,32,33]. The extracts were dried under a flow of N_2_ and stored at −20 °C until analysis. The dried extract residue was dissolved in hexane/2-propanol/H_2_O (718:271:10, *v*/*v*/*v*) injection solvent, filtered through a 0.45 μm, 4 mm diameter True^TM^ Regenerated Cellulose syringe filter (Grace Davison) prior to injection and analysis. Extracted IPLs from the enrichment slurries were analyzed by high-performance liquid chromatography–electrospray ionization–ion trap mass spectrometry (HPLC-ESI/IT/MS) according to [34], with some modifications [35]. IPL abundance was assessed by integrating the HPLC-ESI/IT/MS mass chromatogram for each individual identified IPL, and areas were summed per IPL head group and reported per gram of slurry, dry weight.

### 2.10. Statistical Analysis

Redundancy analysis was performed as implemented in the CANOCO 5 software package (Biometris, Wageningen, The Netherlands) in order to assess to what extent experimental variables influenced the microbial community composition. The experimental variables tested were the incubation temperature, total concentrations of sulfate, butyrate, acetate, and methane consumed/produced by the end of the incubations. A Monte Carlo permutation test based on 499 random permutations was used to determine which of the experimental variables significantly contributed to the observed variance in the composition of microbial communities at the order (for bacteria) and family level (for archaea). Orders and families of at least 5% relative abundance in any sample were included in the analysis. The community structure was visualized via ordination triplots with scaling focused on intersample differences. Correlations between bacterial and archaeal groups and experimental parameters were determined by means of the two-tailed Spearman’s rank-order correlation test using the statistical software SPSS Statistics (IBM SPSS Statistics, Version 22, IBM Corp., Armonk, NY, USA). A statistical significance level of 5% was applied. Multivariate canonical correspondence analysis (CCA) [36] was performed using R statistical analysis software to find phylogenetic groups and IPLs that correlate with each other or with particular incubation conditions.

## 3. Results

### 3.1. Sampling Site Geochemistry

Zones were defined based on sulfate and methane concentrations determined using pore water extracted from sediment during the sampling cruise. The sulfate concentration decreased from 18.5 mM at 15 cm of the core to a low background value at 170 cm. Methane increased steeply with depth below 120 cm and reached a plateau of 2 mM at 225 cm. The sediment core was divided into three pieces representing the sulfate zone (SZ) (15–120 cm), the sulfate–methane transition zone (SMTZ) (120–170 cm), and the methane zone (MZ) (170–300 cm) (Appendix A).

### 3.2. Slurry Incubations

#### 3.2.1. Sulfate Zone Sediment Slurries

Slurries with sediment from the sulfate zone were incubated for 514 days at 25 °C and 10 °C (Figure 1). Total amounts of the butyrate and sulfate consumed and acetate, sulfide, and methane produced in all slurries are listed in Appendix A. Conversion of butyrate in sulfidogenic slurries started after 12 days of incubation (Figure 1A,B). Repeated additions of butyrate and sulfate caused a steady increase in acetate and sulfide. Methane formation was observed after 309 days at 25 °C and after 260 days at 10 °C, and methane increased with time (Figure 1A,B).

In the methanogenic slurries, where no additional sulfate was added, about 1 mM sulfate was measured at the start of the incubation, which presumably originated from the sediment (Figure 1C,D). In the first 40 days of incubation at 25 °C, butyrate conversion coupled to sulfate reduction occurred. Methane formation started on day 50, after sulfate had been depleted. Repeated additions of butyrate yielded high methane and acetate accumulation. Acetate was gradually consumed after day 203 at 25 °C (Figure 1C). Butyrate conversion in the 10 °C slurry was rather slow (Figure 1D). Trace amounts of sulfate that originated from the sediment depleted within 70 days, and methane formation started on day 122. The methane concentration in the 10 °C slurry at the end of the incubation period was three times lower than in the 25 °C slurry.

#### 3.2.2. Sulfate–Methane Transition Zone Sediment Slurries

Slurries of SMTZ sediments with low (3 mM) and high (20 mM) sulfate were incubated for 571 days at 25 °C and 10 °C (Figure 2). Butyrate conversion in high-sulfate slurries coupled to sulfate reduction at both temperatures (Figure 2A,B). Acetate and sulfide concentrations increased by repeated butyrate and sulfate feeds. Methane production started on day 120 at 25 °C (Figure 2A) and on day 229 at 10 °C (Figure 2B), and increased slowly. The amount of methane ranged between 8 and 48 µmole at both temperatures until 500 days. However, it reached to around 4200 µmole at day 571 in slurry at 25 °C. 

In the low-sulfate slurries, the total amount of butyrate converted in the 25 °C slurry was slightly higher than in the 10 °C slurry (Figure 2C,D). Twice as much methane concentration was measured in the 25 °C slurry by the end of incubation as compared with the 10 °C slurry. Acetate accumulated in the 10 °C slurry (Figure 2D), whereas it was initially formed and consumed after 300 days of incubation in the 25 °C slurry (Figure 2C).

#### 3.2.3. Methane Zone Sediment Slurries

Methane zone sediment slurries were incubated for 570 days at 25 °C and 10 °C with and without the addition of sulfate (20 mM) (Figure 3). In the 25 °C slurries, conversion of butyrate started directly (Figure 3A,C). Repeated additions of butyrate and/or sulfate led to a steady increase in acetate and sulfide concentrations. Methane formation in the sulfidogenic slurry was observed after 90 days at 25 °C (Figure 3A), and methane concentration increased with time, whereas no methane was detected at 10 °C (Figure 3B) during the whole incubation period.

Sulfate reduction and methanogenesis co-occurred in the sulfidogenic slurry incubated at 25 °C (Figure 3A). The fastest butyrate conversion was observed in the methanogenic slurry at 25 °C (Figure 3C). Trace amounts of sulfate detected in the slurry at the beginning of incubation were reduced during butyrate conversion within the first 40 days. Methane was detected at day 64 and increased rapidly due to fast conversion of butyrate. Hence, acetate and methane amounts increased steeply within 200 days. A steady acetate consumption occurred after day 214. Even though acetate was produced after butyrate feeds, acetate was consumed again almost completely after 500 days of incubation. Since the methane pressure in these bottles was very high, at day 277, excess headspace gas had to be exhausted (Figure 3C). Butyrate conversion in the methanogenic slurry at 10 °C proceeded much slower (Figure 3D). Methane production started at day 200, and methane concentrations increased gradually. In total, 11 times less methane was produced in the methanogenic slurry incubated at 10 °C compared with the one incubated at 25 °C within 570 days.

### 3.3. Bacterial and Archaeal Community Composition

The long-term incubation of the sediment slurries with butyrate in the presence and absence of sulfate resulted in the development of different microbial communities at the end of incubation. PCR-amplified partial 16S rRNA gene fragments obtained from the last sampling time of all slurries were sequenced. After filtering and trimming, between 3202 and 18,687 high-quality sequences were detected per sample (Appendix A), and these clustered in 62–133 operational taxonomic units (OTUs) per sample. OTUs were classified into 33 phyla, with 96% of the OTUs belonging to 6 phyla, namely, *Proteobacteria* (45.9%), *Chloroflexi* (23.6%), *Firmicutes* (17.7%), *Bacteroidetes* (5.3%), *Spirochaetes* (2.7%), and Candidate division OP9 (1.1%). Different phylotypes were abundant in libraries from the different zones. In sulfate and methane zone sediment libraries, sequences belonging to *Gammaproteobacteria* (64% and 54%, respectively) were dominant, whereas in the SMTZ sediment libraries, sequences related to *Desulfobacteraceae* (79%) belonging to the class *Deltaproteobacteria* were highly detected (Appendix A). Within the *Proteobacteria* phylum, *Desulfobacteraceae* was the main family, containing 61% of the reads. A similar dominance of the *Anaerolineaceae* family with 89.2% reads was observed in the *Chloroflexi* phylum. In the *Firmicutes* phylum, the *Syntrophomonadaceae* family covered 39.9% of the reads, while 50.9% of the reads belonged to uncultured *Firmicutes*. All slurries were dominated by *Desulfobacteraceae, Anaerolineaceae, Syntrophomonadaceae,* and uncultured *Clostridiales*. *Anaerolineales* was observed to be negatively correlated to the other two dominant orders, *Desulfobacterales* and *Clostridiales,* but positively correlated to all environmental parameters, with a strong correlation to temperature (*p* < 0.01), butyrate (*p* < 0.01), and acetate (*p* < 0.05) (Figure 4B and Appendix A). The relative abundance of *Desulfobacterales* (OTU 27) was associated with sulfate concentration and independent of the temperature (Figure 4B and Appendix A). Despite the overall dominance of these families, the relative abundances of each family varied in the different slurry samples. The most abundant OTUs belonged to *Desulfobacterium* in the phyla *Proteobacteria*, uncultured *Anaerolineaceae* in the phylum *Chloroflexi, Syntrophomonas* in the phyla *Firmicutes,* uncultured *Desulfobacteraceae* in the phyla *Proteobacteria,* and uncultured *Clostridiales* in the phylum *Firmicutes* (Figure 4A).

OTU numbers and corresponding taxa are as follows: (1) Bacteria–Other, (2) Actinobacteria-OPB41, (3) Bacteroidetes-BD2-2, (4) Bacteroidetes-Bacteroidales, (5) Bacteroidetes-Cytophagales, (6) Bacteroidetes-Flavobacteriales, (7) Bacteroidetes-SB-1-uncultured, (8) Bacteroidetes-B-5-other, (9) Bacteroidetes-Sphingobacteriales, (10) Bacteroidetes-VC2.1.Bac22, (11) Bacteroidetes-vadinHA17-other, (12) Bacteroidetes-vadinHA17-uncultured bacterium, (13) Candidate division OP9-uncultured bacterium, (14) Chloroflexi-Anaerolineae-other, (15) Chloroflexi-Anaerolineales, (16) Chloroflexi-GIF9-uncultured bacterium, (17) Chloroflexi-MSBL5-uncultured bacterium, (18) Chloroflexi-vadinBA26-uncultured bacterium, (19) Cyanobacteria-Chloroplast-uncultured bacterium, (20) Deferribacteres-LCP-89, (21) Elusimicrobia-Lineage_IV-uncultured bacterium, (22) Firmicutes-Clostridiales, (23) Nitrospirae-Nitrospirales, (24) Proteobacteria-Burkholderiales, (25) Proteobacteria-Hydrogenophilales, (26) Proteobacteria-Desulfarculales, (27) Proteobacteria-Desulfobacterales, (28) Proteobacteria-Desulfovibrionales, (29) Proteobacteria-Desulfuromonadales, (30) Proteobacteria-Sva0485, (31) Proteobacteria-Campylobacterales, (32) Proteobacteria-Gammaproteobacteria-other, (33) Proteobacteria-Alteromonadales, (34) Proteobacteria-Pseudomonadales, (35) Proteobacteria-Thiotrichales, (36) Spirochaetes-LK-44f-uncultured bacterium, (37) Spirochaetes-MSBL2-uncultured bacterium, (38) Spirochaetes-PBS-18-other, (39) Spirochaetes-Spirochaetaceae, (40) Tenericutes-Acholeplasmatales.

The number of reads per incubated slurry for archaeal sequences varied from 9120 to 136,909 (Appendix A). In all libraries from slurry samples, the highest average percentage of 16S rRNA reads for Archaea clustered within the families *Methanomicrobiaceae* (60.8%), *Methanosarcinaceae* (16.3%), and *Methanosaetaeceae* (9.3%) (Appendix A). The fractional abundance of the two aceticlastic methanogenic families *Methanosaetaceae* and *Methanosarcinaceae* did not correlate to each other or other taxonomic groups and environmental parameters (Figure 5A and Appendix A). Both families did not correlate with the hydrogenotrophic methanogenic family *Methanomicrobiaceae*, among which *Methanosarcinaceae* showed significant negative correlation (*p* < 0.05). The relative abundance of *Methanosaetaceae* increased with increasing temperature, but was negatively affected by the presence of sulfate. *Methanosarcinaceae* did not show any significant positive correlation to any of the environmental parameters (Figure 5A and Appendix A). Unclassified *Methanomicrobiales* and EJ-E01 were positively correlated to methane, whereas *Methanomicrobiaceae* was positively correlated only to acetate (Appendix A). This result is consistent with the fact that *Methanomicrobiaceae* had high relative abundance in the slurries in which acetate accumulated to very high concentrations (Figure 5B). *Methanogenium* belonging to the family *Methanomicrobiaceae* was the most dominant genus among the slurries, followed by the genus *Methanosarcina* from the family *Methanosarcinaceae* (Figure 5A). *Methanosaetaeceae* was dominant only in two slurry samples incubated at 25 °C with low and without sulfate, with 43% and 67% of the reads, respectively. These reads belonged to *Methanosaeta* and unclassified *Methanosaetaceae* (Figure 5A).

The intact polar lipid (IPLs) distribution was used as a chemotaxonomic marker. IPL distributions varied substantially over time and differed between the various slurry incubations (Appendix A). Canonical correspondence analysis clustered the observed bacterial OTUs and IPLs into three groups (Appendix A; *with abbreviations explained for the IPL groups*). These groups are as follows: (1) phosphatidylethanolamine (PE), phosphatidic acid (PA), ornithine (OL), phosphatidylglycerol (PG), and phosphatidylinositol (PI) clustered more closely with *Flavobacteriaceae* (BBF), *Cytophagaceae* (BBC1), *Bacteriodetes-*SB-5 (BBS5), *Marinilabiaceae* (BBM), *Actinobacteria*-OPB41 (BAO), *Bacteroidetes*-BD2-2 (BBB), *Bacteroidetes*-VC2.1_Bac22 (BBV), and high-sulfate conditions; (2) monohexose (MH) and glucuronic acid (GlcA) clustered more closely with *Anaerolineaceae* (BCA2), Candidate division OP9 (BCO), Other Bacteria (BO), and low-sulfate conditions; (3) phosphatidylcholine (PC), lyso-phosphatidylcholine (LPC), dihexose (DH), trimethylornithine (TMO), Betaine, and dimethylphosphatidylethanolamine (DMPE) clustered more closely with *Bacteroidetes*-SB-1 (BBS), *Bacteroidetes*-WCHB1-69 (BBW), *Bacteroidetes*-vadinHA1 (BBVH), *Chloroflexi*-GIF9 (BCG), and low-sulfate conditions.

## 4. Discussion

### 4.1. Butyrate Conversion in Aarhus Bay Sediment

Our study shows that butyrate conversion in Aarhus Bay sediments is coupled to both sulfate reduction and methane production. The rapid consumption of sulfate in the slurries suggests that sulfate reduction is the dominant pathway of butyrate conversion in Aarhus Bay sediment. This is most likely due to the fact that sulfate reduction is energetically more favorable than methanogenesis [37]. The accumulation of acetate in all slurries indicated incomplete butyrate conversion (Figure 1, Figure 2 and Figure 3). The decrease in acetate concentration coinciding with methane production suggested the occurrence of methanogenesis in the sulfate zone of Aarhus Bay (Figure 1A,C). This agrees with recent reports of the occurrence of methanogenesis in sulfate-rich marine sediments of Aarhus Bay [20]. 

The butyrate conversion trend in sulfate-free SZ and MZ slurries incubated at 25 °C was similar in terms of early methane production and complete acetate consumption (Figure 1C and Figure 3C). Apparently, syntrophic conversion of butyrate under sulfate-free conditions is possible both in the sulfate and methane zones of Aarhus Bay sediments (Appendix A, reactions 1, 5, 6 and 7).

### 4.2. The Effect of Sulfate Concentration on Butyrate Conversion Dynamics

Significant differences in terms of product formation and consumption in high (20 mM) and low (3 mM) sulfate-amended SMTZ slurries at 25 °C suggest that the sulfate concentration is an important environmental factor in butyrate conversion (Figure 2A,C). Reduced and delayed methane formation in high sulfate-amended slurries possibly indicated sulfide inhibition on methanogenesis [38]. Rapid methane formation after day 523 in high-sulfate-containing slurries might be related to the adaptation capacity of hydrogenotrophic methanogens to the slurry conditions, involvement in syntrophic butyrate conversion, and simultaneous increase of acetate (Figure 2A). This suggests that methanogenesis could still occur despite the ongoing sulfate reduction and the high sulfide level. On the other hand, high methane production with concomitant acetate consumption in low sulfate-amended SMTZ slurries indicated an efficient syntrophic butyrate conversion involving acetate- and hydrogen-dependent sulfate reduction and methanogenesis processes along the incubation (Figure 2C). Apparently, low amounts of sulfate stimulated the whole microbiome, resulting in a fast and dedicated population performing both sulfate reduction and methanogenesis.

### 4.3. Bacterial and Archaeal Community Structure in the Enrichment Slurries

In sulfate-amended slurries, the members of *Desulfobacteraceae*, *Desulfovibrionaceae*, *Desulfobulbaceae*, *Syntrophomonadaceae*, and *Clostridiales* are associated with butyrate conversion. The *Desulfobacterium*, *Desulfonema*, *Desulfosarcina*, *Desulfoarculus* genera belonging to *Desulfobacteraceae* couple complete butyrate oxidation to sulfate reduction [39]. The increase in the relative abundance of these genera in sediment slurries from both SZ and MZ suggested the existence of sulfate reducers in the sulfate, sulfate–methane transition, and methane zones of the sediment. In [40], the authors reported a high absolute abundance of *Desulfobacteraceae* in surface as well as subsurface sediments of Aarhus Bay. The relative abundance of *Syntrophomonas* increased in sulfate-free sediment slurries from the sulfate zone and sulfate-amended sediment slurries from the methane zone. This result showed that the presence of sulfate can decelerate the abundance of *Syntrophomonas* species, but did not inhibit the butyrate conversion by *Syntrophomonas*. Sulfate-reducing bacteria that directly couple butyrate oxidation to sulfate reduction generally grow faster than syntrophic butyrate degraders [18,41]. However, the growth rates of some syntrophic butyrate degraders were reported to be higher than those of some butyrate-oxidizing sulfate reducers [41]. One reason may be that they have similar kinetic properties [42] which allows syntrophic butyrate degraders to occur in sulfate-reducing ecosystems such as marine sediments and aquifers [14,18].

*Anaerolineaceae* became dominant at higher temperatures (25 °C) due to their ability to decompose dead biomass [43] and convert acetate in marine sediment slurries [44]. Considering that the slurries in our study were incubated for a very long period of time, the *Anaerolineaceae* might have scavenged organic compounds from decaying cells [45]. Since *Anaerolineaceae* contain all genes of the acetyl-CoA pathway [46], their involvement in acetate degradation cannot be excluded.

The most abundant IPLs identified in most of the sediment slurries are phosphatidylethanolamine (PE), which are often found in aquatic environments and sediments, and are produced by various groups of microorganisms, such as the sulfate reducers *Desulfosarcina variabilis* and *Desulforhabdus amnigenus* [47], and the bacterial members of ANME-1 [48].

The high abundance of SRB at the end of the slurry incubations and the positive correlation between PE IPLs and high-sulfate condition (Appendix A) might indicate that the occurrence of PE can be linked to the presence of SRB in enrichment slurries. The presence of trimethylornithine (TMO) IPLs in three butyrate-amended slurries from the methane zone (Appendix A) was surprising since these lipids were originally identified in planctomycete isolates from ombrotrophic northern wetlands [49], and later observed in northern wetland peat [50], meso-oligotrophic lakes of Minnesota and Iowa [51], and Yellow Stone National Park hot spring microbial communities [52]. No planctomycete phylotypes were detected in any of the butyrate-amended methane zone-derived slurries, indicating that the observed TMOs were produced by other microbial groups or the relative abundance of planctomycete phylotypes was very low.

*Methanogenium*, belonging to *Methanomicrobiaceae*, dominated all slurry incubations, regardless from which zone the sediment originated and independent of the incubation temperature, except for one slurry (SMTZB7) (Figure 5A). Therefore, *Methanogenium* spp., which are specialized in H_2_ and formate utilization [53], and also known as psychrotrophs, are likely responsible for the consumption of H_2_/formate produced as a result of incomplete conversion of butyrate and may act as syntrophic partners.

The dominance of hydrogenotrophic methanogens in near-surface marine sediments has previously been reported [14,20,54]. The absence of methane in sulfate-rich marine sediments might be due to methane oxidation coupled to methane production, leading to methane cycling in marine surface sediment [20]. The *Methanosaetaceae* dominate only the low-sulfate SMTZ and sulfate-free SZ slurries at 25 °C (Figure 5A). This may be due to the low acetate levels towards the end of the incubation period and the high acetate affinity of *Methanosaetaceae* [55]. Another reason can be the competition with acetate-degrading *Desulfobacteraceae,* which were highly abundant in the original SMTZ (Appendix A). Acetate is mainly consumed by sulfate reducers in marine sediments, and some acetate-degrading sulfate reducers have slightly better growth kinetics than *Methanosaeta* [56]. Similar observations were reported by [57] who speculated that low sulfate may allow *Methanosaetaceae* to compete with sulfate reducers for acetate.

## 5. Conclusions

This study demonstrates that methanogenic archaea and sulfate-reducing bacteria are present and active in sediments of the sulfate zone, the sulfate–methane transition zone, and methane zone of Aarhus Bay and that there is no vertical separation of both groups in the sediment profile. Butyrate conversion could occur under both sulfate-reducing and methanogenic conditions regardless of the incubation temperature and the sediment depth (Figure 6). Conversion of butyrate by syntrophic communities throughout the sediment column suggests that continuous supply of available carbon might stimulate syntrophic butyrate degraders in the sulfate zone. Additionally, SRBs in sulfate-depleted sediment can contribute to the butyrate conversion even in the presence of low sulfate. We suggest that both groups of microbes survive in the sediment and butyrate conversion can proceed via sulfate reduction and methanogenesis simultaneously, though only in conditions of low sulfate. The results indicate that H_2_ and CO_2_ may be major substrates for methanogens, and the members of hydrogenotrophic methanogenic family *Methanomicrobiaceae* were the dominant archaea in the slurries. However, there is only limited competition between SRB and methanogens for acetate. The outcome of this study provides conclusive evidence that marine prokaryotes are metabolically flexible at limiting electron donor and acceptor conditions and the phylogenetic groups that are likely to thrive at different sediment depths.

## Figures and Tables

**Figure 1 microorganisms-08-00606-f001:**
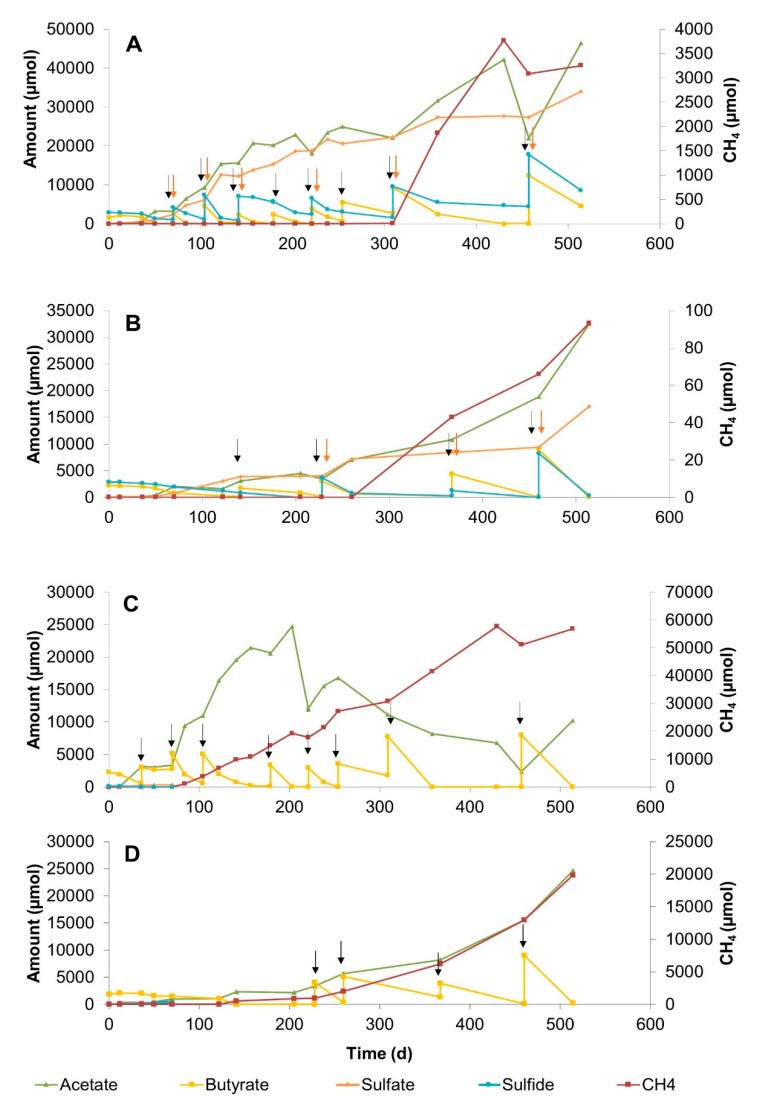
Changes in butyrate, sulfate, acetate, sulfide, and methane concentrations during 514 days of incubation in sediment slurry enrichments constituted of sulfate zone sediment. (**A**) Slurry SZB4, with 20 mM sulfate addition at 25 °C, (**B**) Slurry SZB7, with 20 mM sulfate addition at 10 °C, (**C**) Slurry SZB1, without sulfate addition at 25 °C, (**D**) Slurry SZB5, without sulfate addition at 10 °C. Arrows denote the time points for the additions of sulfate (red) and butyrate (black).

**Figure 2 microorganisms-08-00606-f002:**
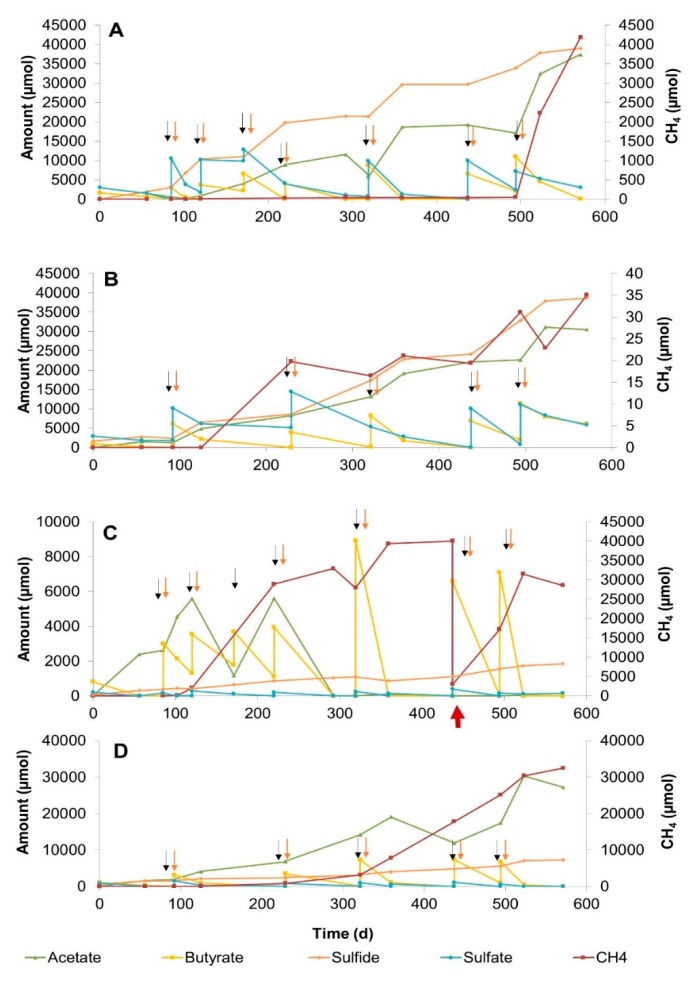
Changes in butyrate, sulfate, acetate, sulfide, and methane concentrations during 571 days of incubation in sediment slurry enrichments constituted of sulfate–methane transition zone sediment. (**A**) Slurry SMTZB3, with 20 mM sulfate addition at 25 °C, (**B**) Slurry SMTZB7, with 20 mM sulfate addition at 10 °C, (**C**) Slurry SMTZB1, with 3mM sulfate addition at 25 °C, (**D**) Slurry SMTZB6, with 3mM sulfate addition at 10 °C. Arrows denote the time points for the additions of sulfate (red) and butyrate (black). The red arrow in Figure 2C indicates the time point that the excess amount of gas was exhausted from the headspace.

**Figure 3 microorganisms-08-00606-f003:**
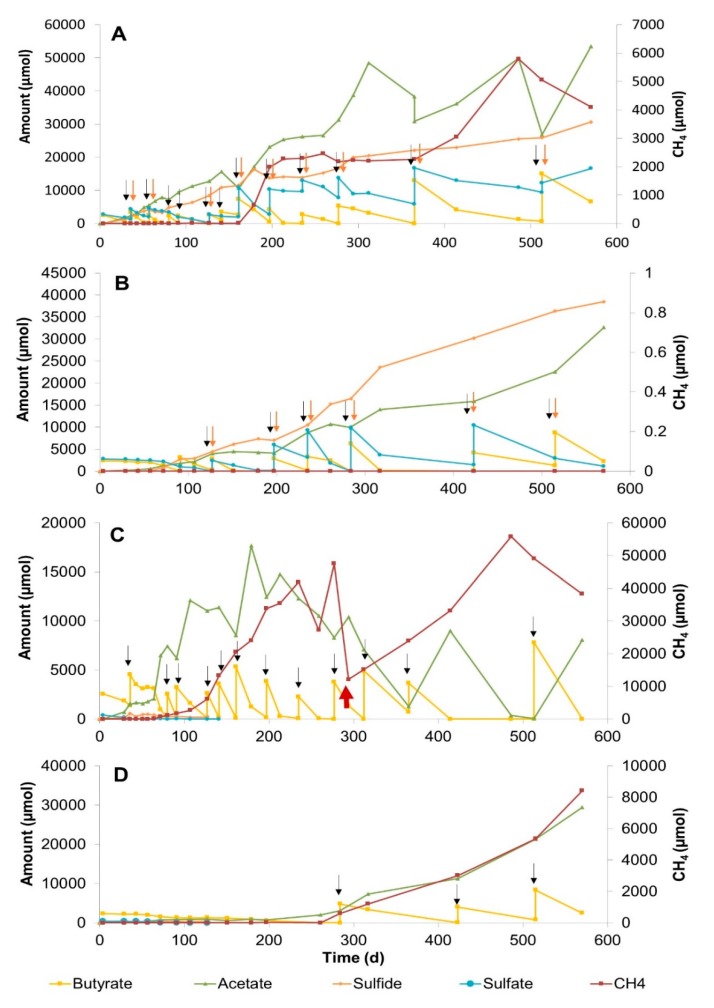
Changes in butyrate, sulfate, acetate, sulfide, and methane concentrations during 570 days of incubation in sediment slurry enrichments constituted of methane zone sediment. (**A**) Slurry MZB3, with 20 mM sulfate addition at 25 °C, (**B**) Slurry MZB2, with 20 mM sulfate addition at 10 °C, (**C**) Slurry MZB5, without sulfate addition at 25 °C. (**D**) Slurry MZB8, without sulfate addition at 10 °C. Arrows denote the time points for the additions of sulfate (red) and butyrate (black). The red arrow in Figure 3C indicates the time point that the excess amount of gas was exhausted from the headspace.

**Figure 4 microorganisms-08-00606-f004:**
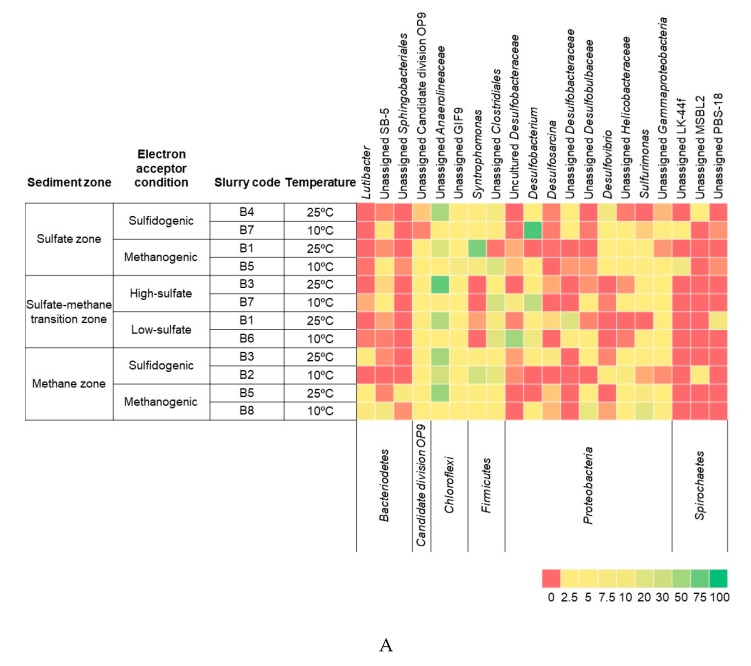
(**A**) The heatmap depicts the relative percentage of the most common (>5 %) bacterial 16S rRNA gene sequences across the 12 slurries analyzed. The heatmap colors represent the relative percentage of the bacterial assignments within each sample. Colors shifted towards dark green indicate higher abundance. Taxonomy is shown at the genus level (unless unassigned) above and at the phylum level below the heatmap. (**B**) Redundancy Analysis Triplot showing relationship between bacterial community composition at order level and environmental parameters. Environmental variables are given as red vectors. Blue vectors represent bacterial orders. Orders were included with a relative abundance of at least 1% in any sample. Vector length gives the variance that can be explained by a particular environmental parameter. Perpendicular distance reflects association, with smaller distances indicating a larger association. Temp: Temperature.

**Figure 5 microorganisms-08-00606-f005:**
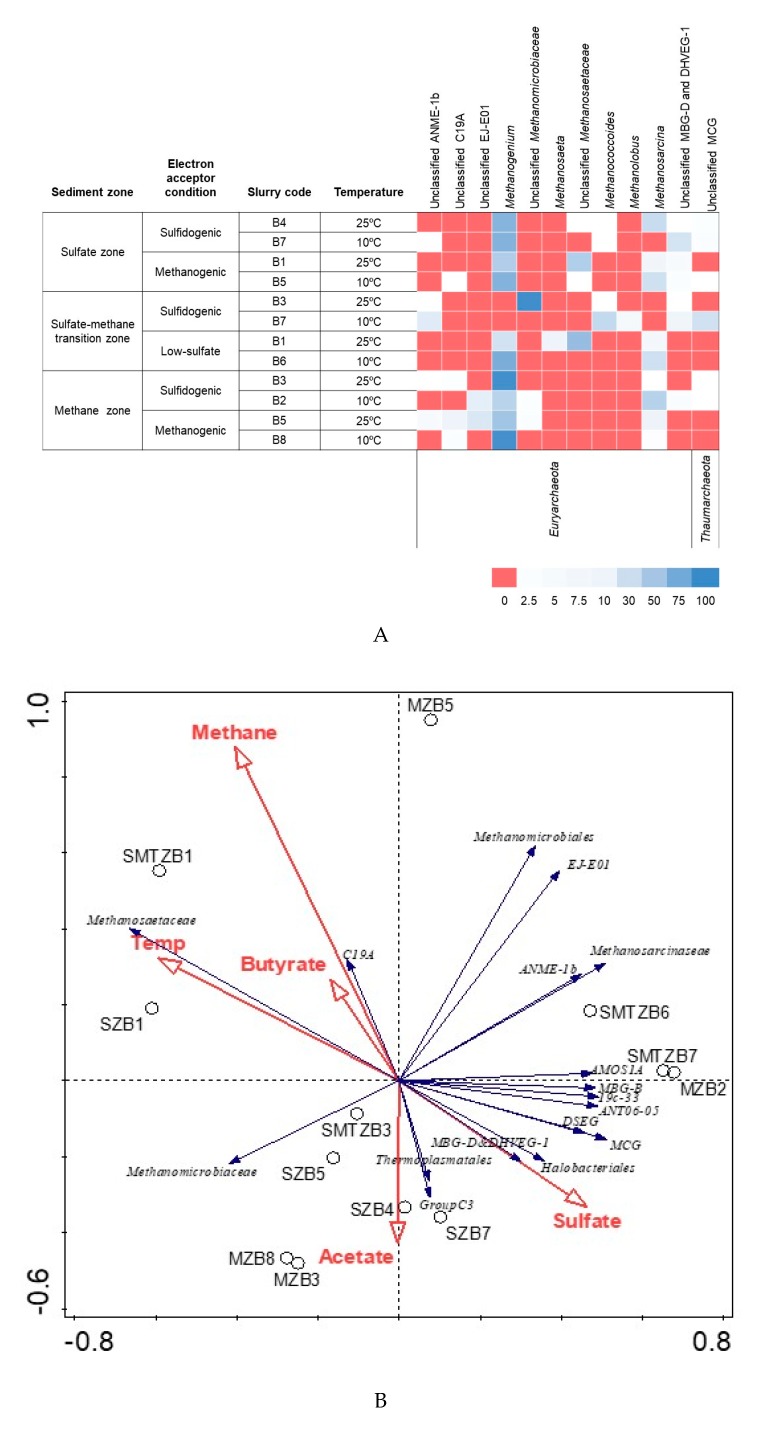
(**A**) The heatmap depicts the relative abundance of the most common (>5 %) archaeal 16S rRNA gene sequences (unless unclassified) across the 12 slurries analyzed. The heatmap colors represent the relative percentage of the archaeal assignments within each sample. Colors shifted towards bright blue indicate higher abundance. (**B**) Redundancy Analysis Triplot showing relationship between Archaeal community composition at family level and environmental parameters. Environmental variables are given as red vectors. Blue vectors represent archaeal families. Families were included with a relative abundance of at least 1% in any sample. Vector length gives the variance that can be explained by a particular environmental parameter. Perpendicular distance reflects association, with smaller distances indicating a larger association. Temp: Temperature.

**Figure 6 microorganisms-08-00606-f006:**
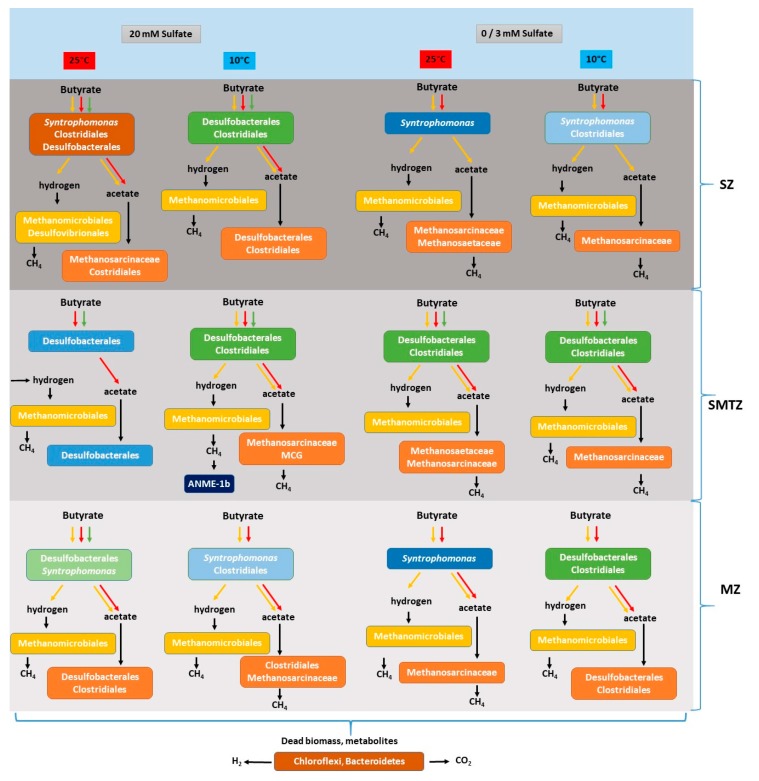
Overview of butyrate conversion and the proposed responsible microbial community at different temperatures and sulfate concentrations in enrichment slurries of sulfate, sulfate–methane transition, and methane zone sediment of Aarhus Bay. Possible butyrate conversion pathways are shown with different colored arrows; red arrows represent incomplete butyrate conversion coupled to sulfate reduction, green arrows represent complete butyrate conversion coupled to sulfate reduction, yellow arrows represent syntrophic butyrate conversion. Horizontal arrow represents the substrates originated from fermentation, decomposition of dead biomass, and/or metabolites.

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
