# Peer review of "Butyrate Conversion by Sulfate-Reducing and Methanogenic Communities from Anoxic Sediments of Aarhus Bay, Denmark"

_microorganisms, 2020, doi:10.3390/microorganisms8040606_

Round 1

Reviewer 1 Report

This is an interesting manuscript that should be useful to many in research. Expression of research is state-of-the-art.  

There are a couple of instances where italics is to be checked.

line 347 contains a phrase in italics. Q. Should this phrase be in italics?

Paragraph starting with line 397. Scientific designations are to be in italics.

line 426 Planctomycete to be in italics?

line 704. Is Chloroflexi to be in italics?

Author Response

Response to Reviewer 1

This is an interesting manuscript that should be useful to many in research. Expression of research is state-of-the-art.  

            Thank you for your compliments on our manuscript

There are a couple of instances where italics is to be checked.

line 347 contains a phrase in italics. Q. Should this phrase be in italics?

We wanted to emphasize that the abbreviations that we used for the Intact Polar Lipids could be found in the supplementary information. If it is not allowed by the desk editor, we will change it. For now we kept it.

Paragraph starting with line 397. Scientific designations are to be in italics.

We are not sure what the reviewer means. As far as we can see, all prokaryotic species/genera/families are presented in italics in this paragraph

line 426 Planctomycete to be in italics?

We checked how this was done by other researchers. In several publications we found the term presented as “planctomycete bacteria” without capital P. Therefore. we changed the text in 3 places from “Planctomycete” to “planctomycete”.

line 704. Is Chloroflexi to be in italics?

In the title of this paper "Chloroflexi" was not presented in italics. Therefore, we kept it as it was presented in the reference.

Reviewer 2 Report

The authors carried out an important research looking into methane production and sulfate reduction via butyrate degradation in Aarhus Bay sediments. They have found some interesting results, which overall show these two processes may occur simultaneously in marine sediments despite sulafte availability. This result would have some important implications in the global carbon cycle, climate change research and biogeochemical sulfur cycling. I am generally happy with how the manuscript has been written and the results have been presented. However, I have some concerns.

My main concerm is how the results were interpreted in the discussion section. The authors claim that butyrate conversion is coupled to both sulfate reduction and methane production in Aarhus Bay sediments (e.g. Lines 369, 450). However, results show that methane generation started after 50 days at the earliest even in the methanogenic slurries. In fact, no methane was observed in the methane zone sediment slurries incubated at 10°C (authors indicated that this is the natural temperature of the Aarhus sediments). So, the discussion should be updated throughout to indicate that the Aarhus sediments have the potential to produce methane whilst sulfate reduction occurs, but it is unlikely under natural conditions.

Minor comments:

Line 134. Please correct the typo in the title.

Line 199. It would be clearer if the section is named experimental set-up or slurry incubations. Then, the sections on sulfate zone slurries etc can be provided.

Figure 1. Colours used are very similar to each other. Please select different ones (preferably just in black and white and denote different data with different markers).

Line 226-228. It looks unconventional and even misleading when the amount of methane is provided as micromole/bottle as this is specific to the experiments in the study. Please just use the amount (e.g. micromole).   

Line 229. Do you mean ‘sulfate-methane transition zone’ samples when you say ‘in the low-sulfate slurries’? The latter is confusing as it may indicate the experimental set-up rather than the sample source.

Line 359. I am confused about the colour-coding. The legend says the red indicates higher abundances but it says 0 in the figure and there are more blue colour in the Archaeal heatmap.

Author Response

Response to Reviewer 2

The authors carried out an important research looking into methane production and sulfate reduction via butyrate degradation in Aarhus Bay sediments. They have found some interesting results, which overall show these two processes may occur simultaneously in marine sediments despite sulafte availability. This result would have some important implications in the global carbon cycle, climate change research and biogeochemical sulfur cycling. I am generally happy with how the manuscript has been written and the results have been presented. However, I have some concerns.

Thank you for your compliments on our manuscript

My main concerm is how the results were interpreted in the discussion section. The authors claim that butyrate conversion is coupled to both sulfate reduction and methane production in Aarhus Bay sediments (e.g. Lines 369, 450). However, results show that methane generation started after 50 days at the earliest even in the methanogenic slurries. In fact, no methane was observed in the methane zone sediment slurries incubated at 10°C (authors indicated that this is the natural temperature of the Aarhus sediments). So, the discussion should be updated throughout to indicate that the Aarhus sediments have the potential to produce methane whilst sulfate reduction occurs, but it is unlikely under natural conditions.

Thank you for your critical evaluation of our discussion.

Therefore, we have updated the discussion indicating that Aarhus Bay sediments have the potential to reduce sulfate and produce methane at the same time, but that under natural conditions of low temperature this is only likely to happen in conditions low in sulfate.

Minor comments:

Line 134. Please correct the typo in the title.

            Sorry, but we did not see a typo in the title “16S rRNA gene amplicon pyrosequencing”

Line 199. It would be clearer if the section is named experimental set-up or slurry incubations. Then, the sections on sulfate zone slurries etc can be provided.

We added section 3.2 “Slurry Incubations” and numbered the zones to 3.2.1; 3.2.2 etc  Thank you

Figure 1. Colours used are very similar to each other. Please select different ones (preferably just in black and white and denote different data with different markers).

We have chosen the same colors throughout figures 1, 2 and 3 in this manuscript and also in a recently published parallel paper in the journal Microorganisms (Propionate Converting Anaerobic Microbial Communities Enriched from Distinct Biogeochemical Zones of Aarhus Bay, Denmark under Sulfidogenic and Methanogenic Conditions). We believe that the in the final high quality version of the figures the colors can be seen well. In addition, there is also the symbol differentiation

Line 226-228. It looks unconventional and even misleading when the amount of methane is provided as micromole/bottle as this is specific to the experiments in the study. Please just use the amount (e.g. micromole).   

We have changed micromole/bottle to micromole. Thank you

Line 229. Do you mean ‘sulfate-methane transition zone’ samples when you say ‘in the low-sulfate slurries’? The latter is confusing as it may indicate the experimental set-up rather than the sample source.

In the slurry incubations from the SMTZ that are presented in paragraph 3.3 we used two concentrations of sulfate (3 and 20 mM) which were termed “low” and “high” sulfate. See line 223 (first line of paragraph 3.3). Moreover, it was also presented in the method section lines 110-112: “10 mM Na-butyrate was used as carbon source with and without 20 mM sulfate in sulfate zone and methane zone slurries, and with 3 mM and 20 mM sulfate for sulfate-methane transition zone slurries”.

As in this paragraph exclusively the results of SMTZ slurry incubations is presented we keep it as it was.

Line 359. I am confused about the colour-coding. The legend says the red indicates higher abundances but it says 0 in the figure and there are more blue colour in the Archaeal heatmap.

The reviewer is right. We changed the legend of both Figure 4A and 5A to:  ‘Colors shifted towards dark green indicate higher abundance’ and ‘Colors shifted towards bright blue indicate higher abundance’ for 5A, respectively. Thank you!